# Structural bioinformatics studies of serotonin, dopamine and norepinephrine transporters and their AlphaFold2 predicted water-soluble QTY variants and uncovering the natural mutations of L->Q, I->T, F->Y and Q->L, T->I and Y->F

Taner Karagöl[1], Alper Karagöl[1], Shuguang Zhang[2]*

1 Istanbul University Istanbul Medical Faculty, Istanbul, Turkey, 2 Laboratory of Molecular Architecture, Media Lab, Massachusetts Institute of Technology, Cambridge, MA, United States of America

⊙ These authors contributed equally to this work.

* taner.karagol@gmail.com

**Data Availability Statement:** The AlphaFold DB [29,30] (https://alphafold.ebi.ac.uk), a database

## Abstract

Monoamine transporters including transporters for serotonin, dopamine, and norepinephrine play key roles in monoaminergic synaptic signaling, involving in the molecular etiology of a wide range of neurological and physiological disorders. Despite being crucial drug targets, the study of transmembrane proteins remains challenging due to their localization within the cell membrane. To address this, we present the structural bioinformatics studies of 7 monoamine transporters and their water-soluble variants designed using the QTY code, by systematically replacing the hydrophobic amino acids leucine (L), valine (V), isoleucine (I) and phenylalanine (F) with hydrophilic amino acids (glutamine (Q), threonine (T) and tyrosine (Y). The resulting QTY variants, despite significant protein transmembrane sequence differences (44.27%-51.85%), showed similar isoelectric points (pI) and molecular weights. While their hydrophobic surfaces significantly reduced, this change resulted in a minimal structural alteration. Quantitatively, Alphafold2 predicted QTY variant structures displayed remarkable similarity with RMSD 0.492Å-1.619Å. Accompanied by the structural similarities of substituted amino acids in the context of 1.5Å electron density maps, our study revealed multiple QTY and reverse QTY variations in genomic databases. We further analyzed their phenotypical and topological characteristics. By extending evolutionary game theory to the molecular foundations of biology, we provided insights into the evolutionary dynamics of chemically distinct alpha-helices, their usage in different chemotherapeutic applications, and open possibilities of diagnostic medicine. Our study rationalizes that QTY variants of monoamine transporters may not only become distinct tools for medical, structural, and evolutionary research, but these transporters may also emerge as contemporary therapeutic targets, providing a new approach to treatment for several conditions.

developed by DeepMind and the European Bioinformatics Institute (EMBL-EBI) at EMBL, is a repository for AlphaFold2 predictions, with over 214 million protein structures. For more detailed information on the statistical analyses, input files and detailed outputs, including the AlphaFold2 calculations, please visit the website: https://github.com/karagol-alper/monoamine-transporters.

**Funding:** The author(s) received no specific funding for this work.

**Competing interests:** Massachusetts Institute of Technology (MIT) filed several patent applications for the QTY code for GPCRs and OH2Laboratories licensed the technology from MIT to work on water-soluble GPCR variants. However, this article does not study GPCRs. SZ is an inventor of the QTY code and has a minor equity of OH2Laboratories. S.Z. founded a startup 511 Therapeutics to generate therapeutic monoclonal antibodies against solute carrier transporters to treat pancreatic cancer. S.Z. has majority equity in 511 Therapeutics. 511 Therapeutics sponsored the study but had no influence and interference in the design of the study, in the data collection, analyses, or interpretation of data and in the writing of the manuscript, or in the decision to publish the results. All other authors have no competing interest. This does not alter our adherence to PLOS ONE policies on sharing data and materials.

## Introduction

Monoamine transporters (MATs) are a class of membrane proteins that play a vital role in regulation of dopamine (DA), norepinephrine (NE), and serotonin (5-hydroxytryptamine [5-HT]) levels in synaptic environments by allowing their reuptake into the presynaptic terminal [1, 2]. Whereas vesicular monoamine transporters (VMATs) include VMAT1 and VMAT2 [3] transports monoamine neurotransmitters into the vesicles, which later released into the synapses. Monoaminergic signaling mediated by these transporters is one of the most fundamental mechanisms of neuronal physiology [1–5].

Reuptake of monoamines terminates the action of the monoaminergic signaling and recycles these neurotransmitters back into the presynaptic neuron [1, 2]. Hence, dysregulations of the plasma membrane monoamine transporters including DAT (dopamine transporter), NET (norepinephrine transporter) and SERT (serotonin transporter) have been implicated in a wide range of conditions such as depression, anxiety disorders, and attention-deficit hyperactivity disorder (ADHD) [1–4]. Given their crucial roles in clinical etiologies, many antidepressant medications work by inhibiting SERT or NET [5, 6].

Both DAT and NET have been widely implicated in two of the most common neurological diseases, Alzheimer's and Parkinson's diseases [2, 4, 5, 7]. Drugs of abuse, including cocaine and amphetamines, also target DAT and NET; and the structural studies have potential to aid in the design of medications that can mitigate addiction-related behavior [4, 5]. Apart from their roles in the central nervous system, serotonin transporter (SERT) is a primary transporter facilitating serotonin uptake from the blood plasma [8]. Several cardiovascular diseases are associated with alterations in plasma serotonin level [8]. For this reason, potential modulators are likely to target SERT in order to overcome these hemodynamic alterations.

Contrary to plasma transporters, vesicular monoamine transporters (VMATs) transport neurotransmitter substrates into secretory vesicles [3]. Expression of the VMATs varies between the two isoforms: VMAT1 is expressed mainly in the peripheral nervous system and neuroendocrine cells, whereas VMAT2 is more prominent in central nervous system [9]. A decrease in VMAT2 expression has been shown to be correlated with an increase in susceptibility to the Parkinson's disease, which may be due to a ratio between the dopamine transporters [10]. Being sequence-related to VMATs, vesicular acetylcholine transporters (VAChT) also transport neurotransmitter substrates and involve vital mechanisms in neurophysiology [11]. Diagnostic difficulties are also persistent among clinical presentations involving dopamine transporter, norepinephrine transporter and serotonin transporter. Idiopathic PD can be differentiated from other forms of Parkinsonism by DAT-specific imaging [7]. Meanwhile, SERT specific neuroimaging provides major guides in clinical management of several conditions [12]. Considering this, studies involving water-soluble variants may drastically advance diagnostic medicine and clinical monitoring [13].

Detailed structural studies have a game-changing potential for investigating efficient medications [4–6]. Despite this effort, structural information on the monoamine transporters is still very limited and many aspects of their molecular mechanisms have remained unknown [6]. This outcome is mainly due to membrane protein nature of monoamine transporters, as studies involving them is a daunting task for researchers [14].

To overcome this big challenge, we presented the simple QTY code designing water-soluble membrane domains [15–17]. The 1.5Å electron density maps show remarkable structural similarities between leucine (L) vs glutamine (Q); isoleucine (I), valine (V) vs threonine (T); and phenylalanine (F) vs tyrosine (Y) [15, 16]. These characteristics led to the implementation of the QTY-code and later the reverse QTY-code to modulate hydrophilicity [15, 16, 18]. The QTY code was previously applied to design a range of detergent-free chemokine receptors and

cytokine receptors, and the expressed and purified water-soluble variants exhibited the predicted characteristics and maintained their ligand-binding activity [15–17, 19, 20].

After AlphaFold2 was publicly released in July 2021, we started using AlphaFold2 to make QTY variant protein structure predictions and achieved improved results, significantly faster [21–24]. In which, the QTY variants of transporter proteins, including potassium channels and glucose transporters, showed a remarkable similarity with their native counterparts [22–24]. Concurrently, reverse QTY variants of Human Serum Albumin successfully designed and demonstrated efficacy in facilitating the release of anti-tumor drugs in mice [18].

Recently, the QTY code designed water-soluble chemokine receptor CXCR4$^{QTY}$ was successfully utilized in biomimetic sensors, called Receptor S-layer Electrical Nano Sensing Array (RESENSA) [13]. Alongside its promising role to develop diagnostics tools, soluble QTY variants of monoamine transporters may have several additional benefits; from research tools of protein engineering; use as water-soluble antigens to accelerate the discovery of effective therapeutic monoclonal antibodies and modulators.

While limited structural knowledge persists for monoamine transporters, a vast amount of information is made available by single nucleotide variants reported for the genes encoding monoamine transporters. The analysis of these variants is essential for the identification of amino acid residues that have specific properties [25]. The QTY code, as a substitution-based design approach, offers distinct advantages for utilizing variant analysis. Another promising research aspect is related to the fact that transporter proteins are conserved from bacteria to mammals [26]. This remarkable conservation makes them ideal subjects for evolutionary studies. For instance, evolutionary profiling potentially allows a better understanding of systematic amino acid substitutions, including the QTY code. Applied to the transmembrane helices of monoamine transporters, QTY code-based studies are likely to provide insights into biological foundations of chemically distinct alpha-helices. Understanding the alpha-helical evolution has a potential to provide crucial information on disease manifestations and sequence-based structural predictions. Moreover, by combining this information with variant based approaches, we analyzed to what extent Darwinian natural selection and mutationally favored biases affected the diversification of QTY-code pairs (L/Q, I/T, V/T, and F/Y).

In this study, we report using the combination of multiple approaches including i) Alphafold2 based structural analysis, ii) variant analysis, and iii) evolutionary conservation studies, to investigate the QTY code design strategy for dopamine transporter (DAT), norepinephrine transporter (NET) and serotonin transporter (SERT). Accompanied by genetic and structural similarities of QTY code pairs, our study uncovered many QTY variants in genomic databases and tracked the phylogenetic presence in homologous sequences.

Our robust statistical analyses provide extensive insights into the evolutionary dynamics of chemically distinct alpha-helices, while also demonstrating the viability of QTY code for a wide range of possible therapeutic applications. QTY variants of monoamine transporters have a potential not only to become distinct tools for structural, medical, and diagnostic research, but also holds promise for accelerating drug discovery in the treatment of various diseases.

## Results and discussions

### Protein sequence alignments and other characteristics

The topological visualizations and predicted sequence features of 7 monoamine transporters indicate that each transporter has a 12-transmembrane (TM) architecture (Fig 1, S1 Fig in S1 File). Their transmembrane alpha-helical segments contain mostly hydrophobic amino acids (Fig 1). Alongside the transmembrane segments. Dopamine transporter (DAT),

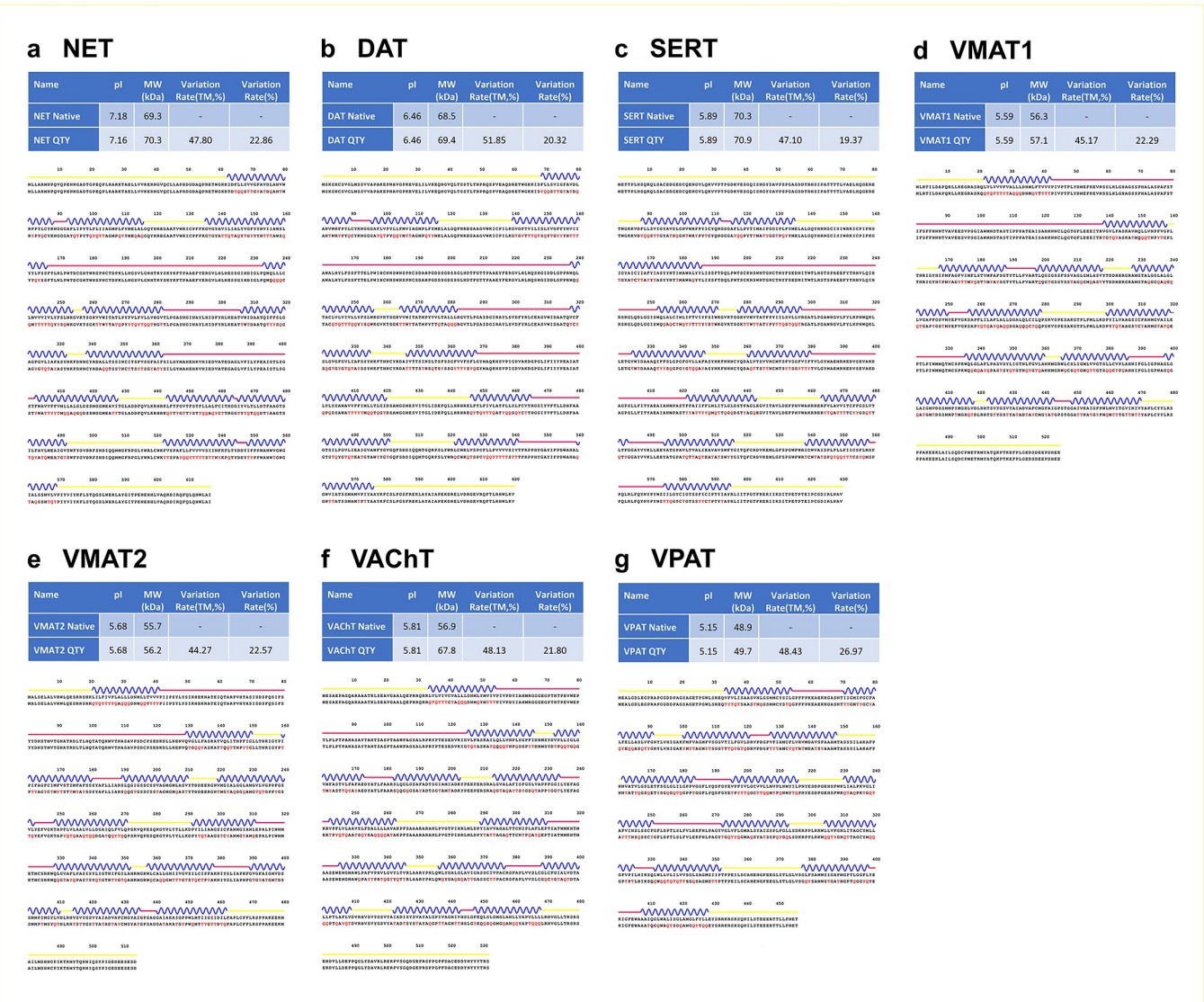

**Fig 1. Sequence and protein alignments of the natural and QTY variants of seven monoamine transporters.** The specific alignments are as follows: **a** NET vs NET^QTY, **b** DAT vs DAT^QTY, **c** SERT vs SERT^QTY, **d** VMAT1 vs VMAT1^QTY, **e** VMAT2 vs VMAT2^QTY, **f** VAChT vs VAChT^QTY, and **g** VPAT vs VPAT^QTY. Molecular weight and isoelectric point (pI) are listed for both the natural and QTY variants. The transmembrane variation percentage and total variation percentage between the natural and QTY variants are also provided. TM alpha-helices (in blue) displayed above the protein sequences and QTY amino acid substitution changes colored in red. Additional color coding is used to distinguish intracellular regions (yellow line), transmembrane helices (blue wave) and extracellular regions (pinkish line). Single-letter abbreviations for the amino acid residues: A, Ala; C, Cys; D, Asp; E, Glu; F, Phe; G, Gly; H, His; I, Ile; K, Lys; L, Leu; M, Met; N, Asn; P, Pro; Q, Gln; R, Arg; S, Ser; T, Thr; V, Val; W, Trp; and Y, Tyr.

norepinephrine transporter (NET) and serotonin transporter (SERT) have a large extracellular loop between TM3 and TM4. For in VMATs, a similar large extracellular loop is located between TM1 and TM2. All seven transporters have N- and C- intracellular segments, these segments exhibit varying lengths, with the largest belonging to SERT (S2 Fig in S1 File).

Application of the QTY code on dopamine transporter, norepinephrine transporter and serotonin transporter resulted in significant substitutions in the transmembrane helices, ranging from 41% to 54% (Table 1). The substitution of the CH3- group (15Da) on leucine (L) and valine (V) with -OH groups (17Da) on glutamine (Q) and threonine (T) results in 2Da change per substitution. Meanwhile, the addition of an OH- group occurs while the substitution of

**Table 1. Characteristics of the 7 monoamine transporters and their water-soluble QTY variants.**

| Name | RMSD | pI | MW (KD) | TM variation (%) | Overall variation (%) |
|---|---|---|---|---|---|
| NET - | - | 7.18 | 69.3 | - | - |
| NET[QTY] | 0.701Å | 7.16 | 70.3 | 47.8 | 22.86 |
| DAT - | - | 6.46 | 68.5 | - | - |
| DAT[QTY] | 1.120Å | 6.46 | 69.4 | 51.85 | 20.32 |
| SERT - | - | 5.89 | 70.3 | - | - |
| SERT[QTY] | 0.492Å | 5.89 | 70.9 | 47.1 | 19.37 |
| VMAT1 - | - | 5.59 | 56.3 | - | - |
| VMAT1[QTY] | 1.084Å | 5.59 | 57.1 | 45.17 | 22.29 |
| VMAT2 - | - | 5.68 | 55.7 | - | - |
| VMAT2[QTY] | 1.515Å | 5.68 | 56.2 | 44.27 | 22.57 |
| VAChT - | - | 5.81 | 56.9 | - | - |
| VAChT[QTY] | 0.603Å | 5.81 | 57.8 | 48.13 | 21.8 |
| VPAT - | - | 5.15 | 48.9 | - | - |
| VPAT[QTY] | 0.696Å | 5.15 | 49.7 | 48.43 | 26.97 |

Residue mean-square distance (RMSD) in Å, Isoelectric focusing (pI), Molecular weight (MW), Transmembrane (TM),— = not applicable. The internal and external loops have no changes, the overall changes are significant, and the TM changes are rather large.

phenylalanine (F) to tyrosine (Y) takes place. The sum of these changes results in a minor increase on the molecular weights of the proteins (Table 1). This minimal change is beneficial since change in molecular weight may affect the protein stability and its clearance rate [27]. Additionally, the QTY substitutions does not introduce any charged residues into the protein, thus with minimal changes of pIs. If charges are introduced, these charges could lead to non-specific interactions [28].

## AlphaFold2 predictions

Experimentally determining the structure of a transmembrane protein is a daunting task, from gene expression to selecting the appropriate detergent for maintaining stability, each step poses a significant challenge [14]. Consequently, the deep-learning Alphafold2 presents itself a promising role in structural biology research, with predictions nearing experimental accuracy [29, 30]. In previous work, we used AlphaFold2 to predict the structures of native and water-soluble QTY variants. These predictions were in good agreement with previously known experimentally-determined native structures [21–24].

In this study, we also utilized AlphaFold2 to predict QTY variants and native transporters, as well as comparing them with an experimentally determined native structure of SERT [31], accessed from RCSB PDB [32]. The predicted Local Distance Difference Test (pLDDT) scores for the predictions also indicated a high degree of confidence in their accuracy, especially in helical regions (S3 Fig in S1 File). However, it's important to note that pLDDT scores utilize the information available in the training data, which may not perfectly reflect specific biological contexts [33]. While the output conformation cannot be reliably controlled, no significant conformational deviation that disrupts direct compressions was identified. All predictions corresponded to the outward-facing conformations.

## Superposition of native transporters and their water-soluble QTY variants

Nature has evolved three chemically distinct types of alpha-helices [16, 34, 35]. The hydrophilic alpha-helices contain mostly polar amino acids D, E, N, Q, K, R, S, T, and Y [16], as

found in water-soluble proteins. On the other hand, hydrophobic alpha-helices are present in transmembrane proteins and contain mostly hydrophobic amino acids [16]. Amphiphilic alpha-helices contain both hydrophobic and hydrophilic amino acid residues. We previously postulated that the tertiary structures of the proteins may remain similar and stable whilst having differing alpha-helical types [15, 16]. This is because alpha-helix is a natural, low-energy conformation for polypeptide chains, and the distribution of hydrophobic and hydrophilic regions can still allow for similar overall topologies [15, 16]. Supporting this, despite a high substitution rate in the transmembrane alpha-helices, the predicted structure of the QTY variant remains similar to its experimentally-determined native structure, quantitatively demonstrated by the root mean square deviation (RMSD). The RMSD value for SERT$^{CryoEM}$ *vs* SERT$^{QTY}$ was 1.619Å (Fig 2). The cryo-EM structure was also superposed with corresponding AlphaFold2 predicted native structures with a RMSD value of 1.541Å. The RMSD result may support the accuracy of AlphaFold2's predictions, as the predicted native structures are in line with the experimentally-determined structures.

Alongside the cryo-EM, the Alphafold2 predicted native SERT structure and its QTY variant was also superposed with 0.492Å, meaning the structures were almost identical in terms of average helical composition and structural features (Table 1, Fig 3). Many monoamine transporters currently do not have experimentally determined structures, as in the case of the majority of the transmembrane proteins. We obtained the structures of these transporters (NET, DAT, VMAT1, VMAT2, VAChT, and VPAT) using AlphaFold2 predictions. The structural similarity between the native and QTY variants was high as demonstrated by the root mean square deviation (RMSD), which ranged from 1.515Å (VMAT2) to 0.603Å (VAChT) (Table 1, Fig 3).

These structural similarities do not have to be correlated with locational features, as evident in the molecular lipophilicity potential (MLP) maps that visualized the hydrophobic patches of the molecular surfaces (Figs 4 and 5). The native structures had a high hydrophobicity content, particularly in their transmembrane alpha-helical segments, resulting them to be insoluble and to aggregate in water without appropriate detergents. By replacing the hydrophobic amino acids L, I, V, and F with hydrophilic ones (Q, T, Y), the hydrophobic patches were significantly reduced (Figs 4 and 5), which makes the variants able to be located within an aqueous solvent. Notably, this shift in hydrophobicity did not adversely affect the alpha-helical structures, which was unexpected prior to a comprehensive series of experiments detailed in our previous publications [15–20].

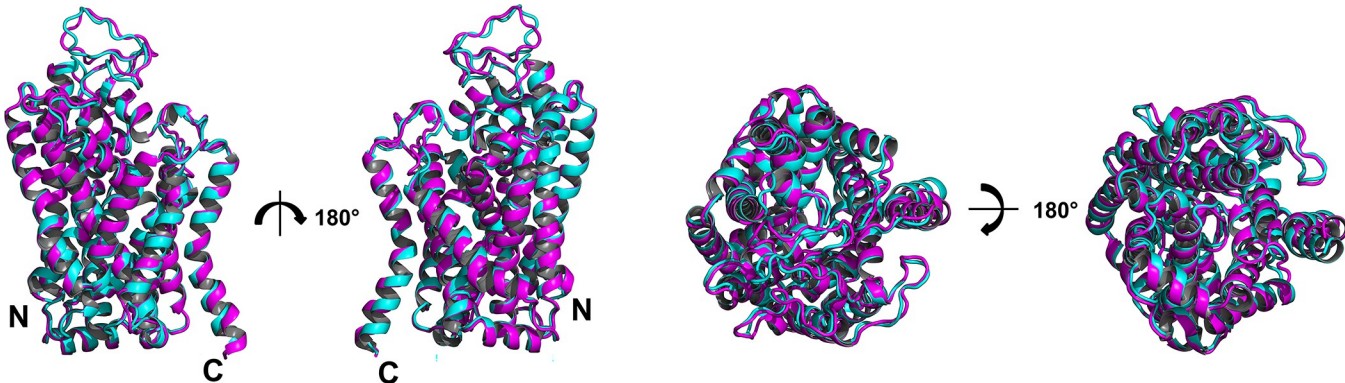

**Fig 2. Superposition of cryo-EM structure SERT$^{CryoEM}$ and AlphaFold2 predicted QTY water-soluble variant SERT$^{QTY}$.** The native SERT (PDB ID: 6VRH) structure is obtained from the Protein Data Bank. N- and C-termini are labelled. The SERT$^{CryoEM}$ (magenta) is superposed with AlphaFold2 predicted water-soluble variant SERT$^{QTY}$ (cyan). The root mean square deviation (RMSD) between the two structures is 1.619Å. N- and C termini and large loops that are not resolved in the experimental structures are removed for clarity of direct comparisons.

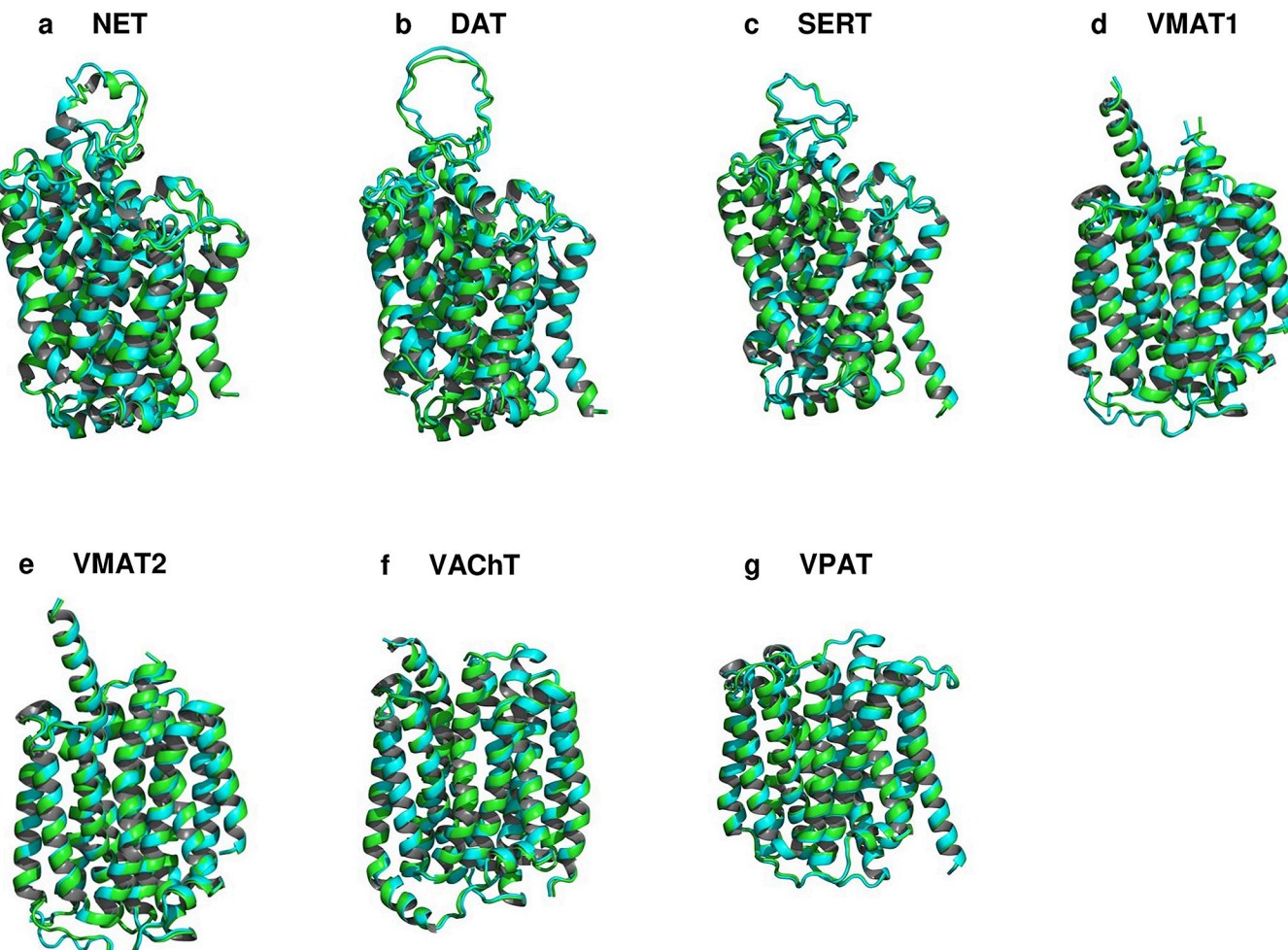

**Fig 3. Superposed 7 native monoamine transporters (MATs) and their QTY variants that were predicted by AlphaFold2.** Native MAT structures are colored in green, and their corresponding water-soluble QTY variants in cyan. RMSD values, expressed in Å (in parentheses), are provided for the following comparisons: **a** NET vs NET^QTY (0.701Å), **b** DAT vs DAT^QTY (1.120Å), **c** SERT vs SERT^QTY (0.492Å), **d** VMAT1 vs VMAT1^QTY (1.084Å), **e** VMAT2 vs VMAT2^QTY (1.515Å), **f** VAChT vs VAChT^QTY (0.603Å), **g** VPAT vs VPAT^QTY (0.696Å). For clarity, N- and C- termini and large loops are deleted. Detailed information can be found in Table 1.

## Natural mutations of L->Q, I->T, F->Y and Q->L, T->I, Y->F in monoamine transporters

The chemical properties of the helical structures are governed by the genetic code, as genetic code's second position is determinant of the chemical nature of amino acids [16, 36, 37]. For instance, **i**) amino acids with U at the second position are hydrophobic (Phe, Leu, Ile, Val, and Met); **ii**) amino acids with C at the second position are less hydrophobic (Pro and Ala), or with a hydroxyl -OH group (Ser and Thr); **iii**) amino acids with A at the second position are hydrophilic and water soluble (Asp, Glu, Asn, Glu, Lys, His and Tyr), and 2 stop codons Ochre (UAA) and Amber (UAG); **iv**) amino acids (Arg and Ser) with G at the second position are water soluble, Cys is partially water-soluble and Gly is achiral and has a hydrogen as the side chain [16, 37]. The stop codon is UGA. In general, pyrimidine U and C at the second position confer hydrophobicity; in contrast, purine A and G at the second position confer hydrophilicity (S4 Fig in S1 File).

**a** **SERT<sup>CryoEM</sup>**

**b** **SERT<sup>QTY</sup>**

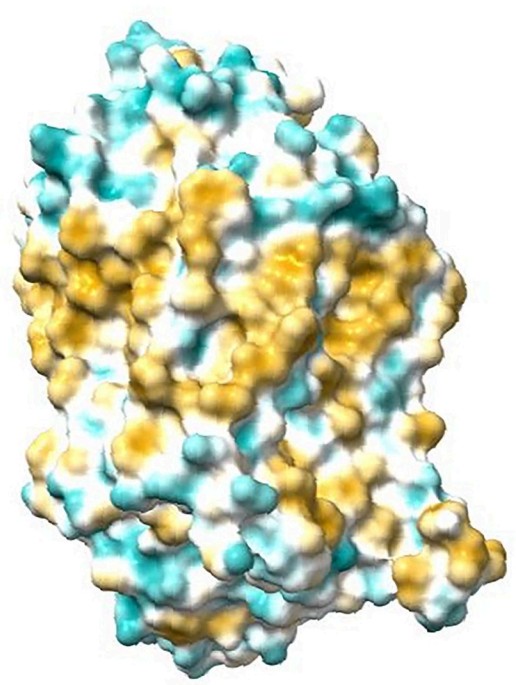
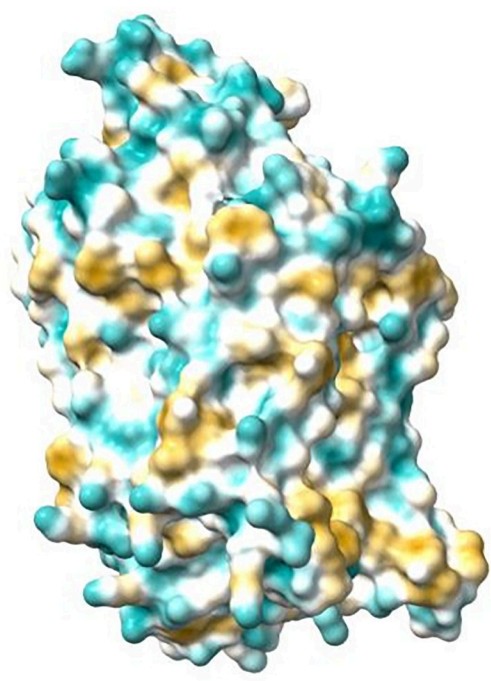

**Fig 4. Hydrophobic surface of the cryo-EM structure and the designed QTY variant of the human serotonin transporter (SERT).** Following the substitution of hydrophobic residues L, I, V, and F with Q, T, and Y, the surfaces showed increased hydrophilicity. The originally hydrophobic surface (brownish) of the native transporter transitioned to a predominantly cyan color, indicating a significant reduction in the hydrophobic surface area along the transmembrane helices for the QTY variant: **a** SERT<sup>CryoEM</sup> vs **b** SERT<sup>QTY</sup>. For clarity, N- and C- termini and large loops are deleted.

In the dopamine transporter, norepinephrine transporter and serotonin transporter, many single nucleotide variations (total 89) resulting in the substitution of L->Q, I->T, F->Y and Q->L, T->I, Y->F were identified from human genomic databases (Fig 6, Tables 2 and 3). Among those, p.Ile136Thr of VMAT1 has a frequency of 0.728. The remaining 88 variations were all rare coding variants with a minor allele frequency (MAF) of less than 0.01. Some I<>T substitutions also had a relatively higher frequency. Namely, p.Thr99Ile ($f$ = 0.0094) and p.Ile549Thr ($f$ = 0.003) of NET, p.Ile168Thr ($f$ = 0.0014) VMAT1, p.Thr68Ile ($f$ = 0.0015) of VMAT2. These high frequencies may reflect their non-disturbing effect on the transporter. As expected, I<>T changes observed more than L<>Q and F<>Y substitutions (69/84 = 82.1%).

Please note that, despite their striking structural similarities, no V->T, nor T->V mutations in the transporters were observed. This is because such changes require at least 2 consecutive nucleotide changes in a single codon, which is exceedingly rare in nature.

The variations all occur in the second position of the genetic code, including transition mutation, i.e., purine to purine (A->G, G->A) and pyrimidine to pyrimidine (C->U, or U->C); or transversion mutation (U->A, U->G, C->A, C->G, A->U, A->C, G->U, G->C). In the case of L->Q, I->T, and F->Y substitutions. **i)** in L (leucine), two codons are CUA and CUG, and in Q (glutamine), two codons are CAA and CAG; in these cases, the second position of U is mutated to A, which is a transversion mutation (U->A). **ii)** In I (isoleucine), three

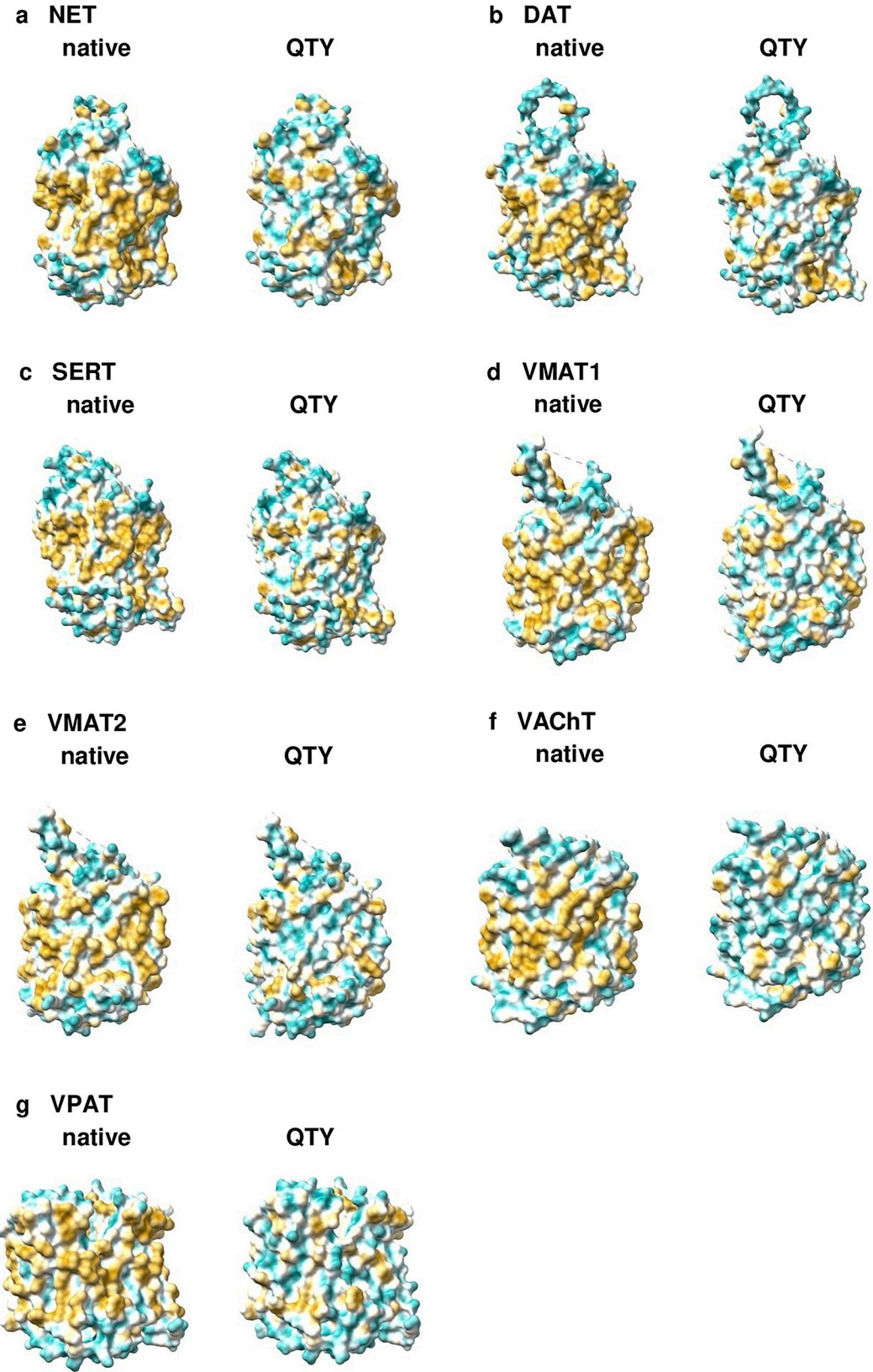

**Fig 5. Hydrophobic surface alterations of seven AlphaFold2-predicted native monoamine transporters and their designed QTY variants.** Following the substitution of hydrophobic residues L, I, V, and F with Q, T, and Y, the surfaces showed increased hydrophilicity. The originally hydrophobic surface (brownish) of the native transporters transitioned to a predominantly cyan color, indicating a significant reduction in the hydrophobic surface area along the transmembrane helices for the QTY variants: **a** NET vs NET$^{QTY}$, **b** DAT vs DAT$^{QTY}$, **c** SERT$^{QTY}$, **d** VMAT1 vs VMAT1$^{QTY}$, **e** VMAT2 vs VMAT2$^{QTY}$, **f** VAChT vs VAChT$^{QTY}$, **g** VPAT vs VPAT$^{QTY}$. For clarity, N- and C- termini and large loops are deleted.

codons are AUU, AUC, and AUA, in T (threonine), four codons are ACU, ACC, ACA, and ACG; which is a transition mutation (U->C). **iii**) In F (phenylalanine), two codons are UUU and UUC, in Y (tyrosine), two codons are UAU and UAC, and the second position of U is mutated to A which is a transversion mutation (U->A). The same logic applies in the case of the reverse QTY mutations of Q->L, T->I, Y->F. Transitions are expected to be more frequent than transversions, with a ratio much higher in coding regions [38]. F<>Y and L<>Q changes that required same mutations (U->A and A->U) observed in similar ratios (11 to 9).

The major mechanism for new mutations is deamination of 5'-methyl C to uracil producing (C->U) or, on the complementary strand, (G->A) [39, 40]. Thus, spontaneous C->T and

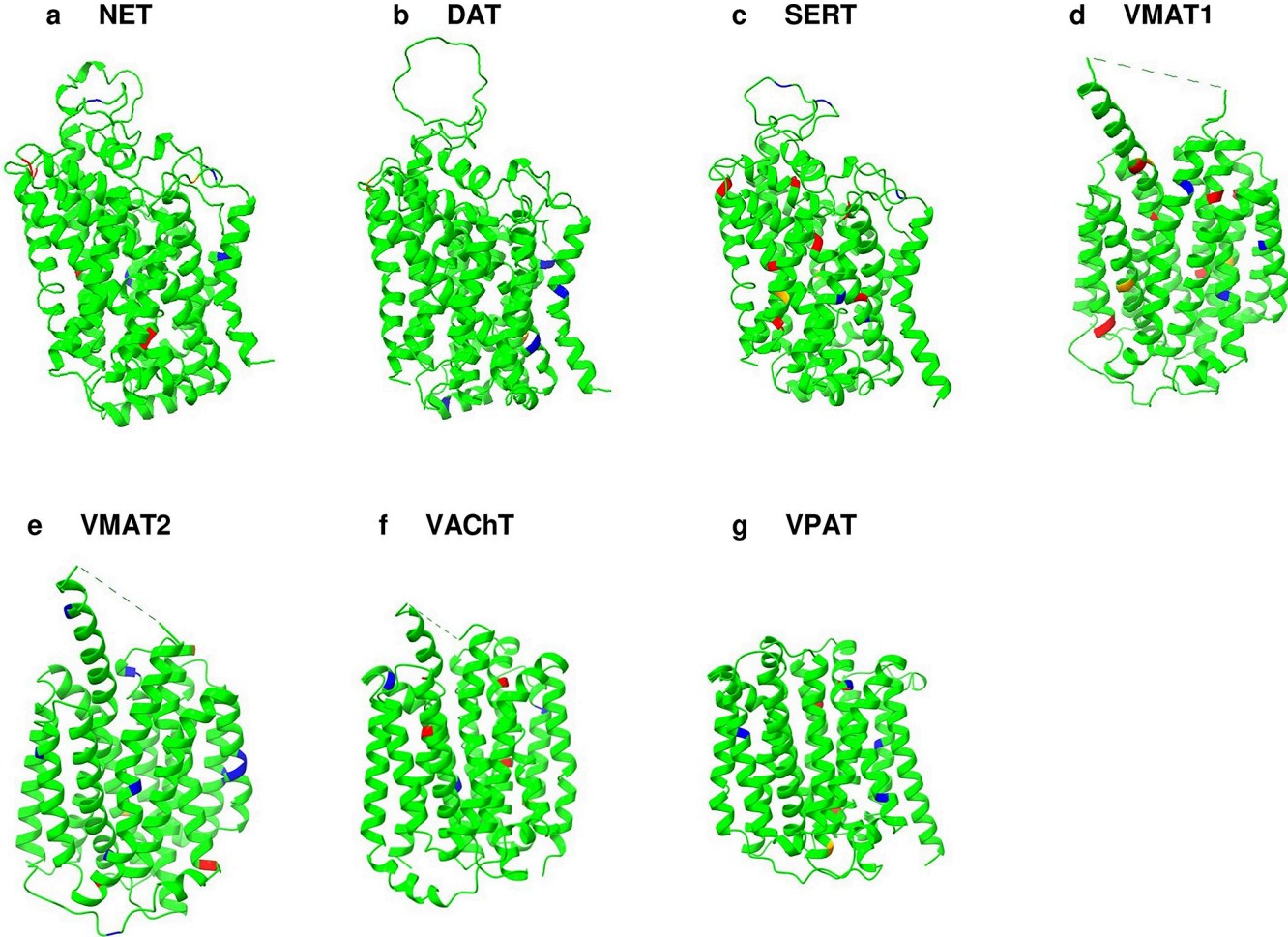

**Fig 6. Natural mutations of QTY-code and rQTY-code.** Native structures are represented in green, while the predicted effects of QTY and reverse QTY mutations are illustrated with colored residues: blue = benign, orange = possibly damaging with low confidence, red = damaging with high confidence. **a** NET, **b** DAT, **c** SERT, **d** VMAT1, **e** VMAT2, **f** VAChT, and **g** VPAT. For clarity, N- and C termini and large loops are deleted.

**Table 2. Natural mutations of L->Q, I->T, F->Y in the 7 monoamine transporters (No V->T mutations.** A single base mutation on the second position of the codons).

| Name | Mutation[1] | 2nd base[2] | Location[3] | Structure[4] | Exposure[5] | Conservation Grade[6] | Predicted Effect[7] | Clinical Significance[8] |
|---|---|---|---|---|---|---|---|---|
| NET | L284Q | U->A | ECL3 | loop | Exposed (F[9]) | 9 | damaging | - |
| | I230T | U->C | ECL2 | loop | Exposed | 4 | ? damaging | - |
| | I428T | U->C | TM8 | α-helix | Buried | 5 | damaging | - |
| | I537T | U->C | TM11 | α-helix | Exposed | 4 | benign | - |
| | I549T | U->C | ECL6 | β-strand | Exposed | 2 | benign | - |
| | F609Y | U->A | Intracellular | 3/10-helix | - | 8 | damaging | Benign |
| DAT | I291T | U->C | ECL3 | α-helix | Exposed | 1 | benign | - |
| | I490T | U->C | TM10 | α-helix | Exposed | 4 | ? damaging | - |
| | I540T | U->C | TM11 | α-helix | Exposed | 4 | benign | Uncertain (P[11]) |
| | F143Y | U->A | TM3 | α-helix | Exposed | 3 | benign | - |
| SERT | L434Q | U->A | TM8 | α-helix | Buried (S[10]) | 9 | damaging | - |
| | I123T | U->C | TM2 | α-helix | Exposed | 5 | ? damaging | - |
| | I165T | U->C | TM3 | α-helix | Buried | 7 | benign | - |
| | I168T | U->C | TM3 | α-helix | Buried | 7 | damaging | - |
| | I218T | U->C | ECL2 | loop | Exposed | 1 | benign | - |
| | I266T | U->C | TM4 | α-helix | Buried | 7 | ? damaging | - |
| | I270T | U->C | TM4 | α-helix | Buried | 7 | damaging | - |
| | I408T | U->C | ECL4 | 5-helix | Buried | 8 | damaging | - |
| | I625T | U->C | Intracellular | C-coil | - | - | benign | - |
| VMAT1 | L49Q | U->A | ECL1 | α-helix | Buried | 8 | damaging | - |
| | I136T | U->C | ECL1 | α-helix | Buried | 8 | benign | - |
| | I168T | U->C | TM3 | α-helix | Exposed | 5 | damaging | - |
| | I202T | U->C | TM4 | α-helix | Exposed | 5 | ? damaging | Uncertain (Inborn) |
| | I309T | U->C | TM7 | α-helix | Buried | 8 | benign | - |
| | I393T | U->C | ECL5 | α-helix | Exposed | 7 | damaging | - |
| | I403T | U->C | TM10 | α-helix | Buried | 8 | ? damaging | - |
| | I490T | U->C | Intracellular | α-helix | - | 7 | damaging | - |
| VMAT2 | L12Q | U->A | Intracellular | α-helix | - | 5 | damaging | - |
| | I149T | U->C | TM2 | α-helix | Buried | 8 | benign | - |
| | I174T | U->C | TM3 | α-helix | Exposed | 4 | benign | - |
| | I223T | U->C | TM5 | α-helix | Exposed | 7 | ? damaging | - |
| | I339T | U->C | TM8 | α-helix | Exposed | 6 | benign | - |
| | I366T | U->C | TM9 | α-helix | Exposed | 4 | benign | - |
| | I458T | U->C | TM12 | α-helix | Exposed | 4 | benign | - |
| | I459T | U->C | TM12 | α-helix | Exposed | 5 | benign | Uncertain |
| | I500T | U->C | Intracellular | C-coil | - | 1 | benign | - |
| | I505T | U->C | Intracellular | C-coil | - | 4 | benign | Uncertain |
| | F148Y | U->A | TM2 | α-helix | Exposed | 6 | damaging | - |
| VAChT | L49Q | U->A | TM1 | α-helix | Buried (S[10]) | 9 | damaging | - |
| | L218Q | U->A | TM5 | α-helix | Buried (S[10]) | 9 | benign | - |
| | I56T | U->C | ECL1 | α-helix | Buried (S[10]) | 9 | damaging | - |
| | I373T | U->C | TM9 | α-helix | Exposed | 3 | benign | - |
| | I394T | U->C | TM10 | α-helix | Buried | 8 | damaging | - |
| VPAT | L258Q | U->A | ECL4 | α-helix | - | 8 | damaging | - |
| | L294Q | U->A | ICL4 | α-helix | - | 4 | damaging | - |
| | I240T | U->C | TM7 | α-helix | - | 5 | benign | Uncertain (Inborn) |

(*Continued*)

**Table 2.** (Continued)

| Name | Mutation[1] | 2nd base[2] | Location[3] | Structure[4] | Exposure[5] | Conservation Grade[6] | Predicted Effect[7] | Clinical Significance[8] |
|---|---|---|---|---|---|---|---|---|
| | F277Y | U->A | TM8 | α-helix | - | 8 | damaging | - |

[1]Protein consequence of the mutation according to HGVS numbering.

[2]The second base of the residue codon for the corresponding mutation.

[3]Topological localizations of the mutations according to monoamine transporter molecular architecture (TM = Transmembrane, ECL = Extracellular loop, IM = Intramembrane, ICL = Intracellular loop). The topological information of the mature protein obtained from Uniprot.

[4]Secondary structure of the corresponding residue, calculated from the determined models of native transporters available in the AlphaFold Database.

[5]Residue exposure according to the NACSES algorithm, predicted by ConSurf server

[6]Evolutionary conservation grade of the residue predicted by ConSurf server; 1 to 9, in order of increasing conservation (1 = Variable, 5 = Average, 9 = Conserved).

[7]Variant effect predicted by Polyphen-2. Benign = predicted to be benign with high confidence;? damaging = possibly damaging, predicted to be damaging with low confidence; damaging = probably damaging: predicted to be damaging with high confidence.

[8]Based on ClinVar's April 09, 2023 release.

[9]A functional residue (exposed and highly conserved) predicted by ConSurf Server.

[10]A structural residue (buried and highly conserved) predicted by ConSurf Server.

[11]P = Parkinsonism

G->A changes are more frequent than T->C and A->G substitutions in Escherichia coli and mammalian cells [40, 41]. However, the T->I substitutions were seen less than I>T changes (34 to 35). At the first stance this may seem counterintuitive, but this observation makes complete biological sense since there are more I (nonpolar) than T (polar) amino acids in the studied transporters (267 to 203). Observed mutations passively reflect mutational properties of the genetic code, as distributions are roughly in line with expectations. In other words, instead of the distributions from natural selection acting and producing observational biases, the codon characteristics may more affected the studied variant distributions. It is also possible that structural similarities of QTY code resulted in the presentation we see, as fatal mutations would produce sample biases [38]. In this study, we did not systematically examine other substitutions since it is beyond the scope of this study.

## Topological analysis of the natural QTY-code and rQTY-code variants

A total of 89 variants, 47 QTY and 42 reverse QTY (rQTY), were identified in the transporter genes, which were located at various positions within the transporter protein (Fig 6). Topological localization of the QTY mutations revealed that 29 out of 47 (61.7%) of the variations were located in the transmembrane (TM) regions. This finding can be attributed to the likely presence of polar L, I, and F amino acids within the TM helices. For the predicted effects of these variations, predictions from Polyphen-2 [42] defined 14 of those as benign (14/29 = 48.2%), 6 as "possibly" damaging with low confidence (6/29 = 20.6%), and 9 as probably damaging (9/29 = 31%). Notably, regardless of their location, more than half of the natural QTY mutations were predicted to be benign (Table 3). Per-residue secondary structure assignment from AlphaFold2 determined models showed that 40 mutations belong to a helical structure, and 16 of those were benign (Table 3).

On the other hand, more than half of the 42 natural mutations of the reverse QTY (Q->L, T->V/I, Y->F) examined in this study were found outside the TM regions (25/42 = 59.5%). In detail, 17 of the rQTY mutations were found in the extracellular regions, and 8 in the cytoplasmic regions. Outside the TM regions, 13 mutations were predicted to be benign (13/25 = 52%), 7 as "possibly" damaging with low confidence (7/25 = 28%), and 5 as probably

**Table 3. Natural mutations of Q->L, T->I, Y->F in the 7 monoamine transporters (No T->V mutations.** A single base mutation on the second position of the codons).

| Name | Mutation[1] | 2nd base[2] | Location[3] | Structure[4] | Exposure[5] | Conservation Grade[6] | Predicted Effect[7] | Clinical Significance[8] |
|---|---|---|---|---|---|---|---|---|
| NET | T99I | C->U | TM2 | α-helix | Exposed | 3 | benign | Benign |
| | T168I | C->U | ECL2 | loop | Exposed | 5 | damaging | - |
| | T186I | C->U | ECL2 | loop | Exposed | 4 | benign | - |
| | T353I | C->U | TM7 | α-helix | Buried (S[10]) | 9 | damaging | - |
| DAT | Q509L | A->U | ICL5 | α-helix | Exposed | 4 | benign | - |
| | T46I | C->U | TM9 | N-coil | - | 5 | benign | - |
| | T286I | C->U | ECL3 | 3/10-helix | Exposed (F[9]) | 8 | ? damaging | - |
| | T566I | C->U | TM12 | α-helix | Exposed | 4 | benign | - |
| SERT | T67I | C->U | Intracellular | N-coil | - | 5 | benign | - |
| | T178I | C->U | TM3 | α-helix | Buried | 7 | damaging | - |
| | T221I | C->U | ECL2 | loop | Exposed | 3 | benign | - |
| | T421I | C->U | ECL4 | α-helix | Exposed | 6 | damaging | - |
| | T472I | C->U | TM9 | α-helix | Exposed | 2 | benign | - |
| | T480I | C->U | TM9 | 3/10-helix | Buried | 6 | damaging | - |
| | Y134F | A->U | TM2 | α-helix | Buried | 6 | damaging | - |
| | Y570F | A->U | ECL6 | loop | Buried | 7 | benign | - |
| VMAT1 | T39I | C->U | TM1 | 3/10-helix | Buried | 8 | damaging | - |
| | T47I | C->U | ECL1 | α-helix | Exposed | 7 | ? damaging | - |
| | T322I | C->U | ECL4 | α-helix | Buried | 8 | damaging | - |
| | T353I | C->U | TM8 | α-helix | Buried | 8 | damaging | - |
| | T464I | C->U | TM12 | α-helix | Buried | 7 | benign | - |
| | T499I | C->U | Intracellular | C-coil | - | 3 | ? damaging | - |
| VMAT2 | Q280L | A->U | ICL3 | loop | Exposed | 3 | benign | - |
| | Q329L | A->U | TM8 | 3/10-helix | Exposed | 7 | benign | - |
| | T68I | C->U | ECL1 | loop | - | 4 | benign | Benign |
| | T97I | C->U | ECL1 | loop | - | 1 | benign | - |
| | T103I | C->U | ECL1 | loop | - | 3 | benign | - |
| | T140I | C->U | TM2 | α-helix | Exposed | 5 | benign | Uncertain |
| | T153I | C->U | ICL1 | α-helix | Buried (S[10]) | 9 | damaging | - |
| | T322I | C->U | ECL4 | α-helix | Exposed | 6 | ? damaging | - |
| | Y293F | A->U | TM7 | α-helix | Exposed | 7 | damaging | - |
| VAChT | T81I | C->U | ECL1 | loop | - | 6 | benign | - |
| | T97I | C->U | ECL1 | loop | - | 2 | benign | - |
| | T311I | C->U | ECL4 | α-helix | Buried | 8 | damaging | - |
| | Y236F | A->U | ECL3 | α-helix | Exposed (F[9]) | 9 | benign | - |
| | Y359F | A->U | TM9 | α-helix | Buried | 5 | ? damaging | Uncertain |
| VPAT | T114I | C->U | TM3 | α-helix | - | 8 | benign | - |
| | T257I | C->U | ECL4 | α-helix | - | 9 | benign | - |
| | T313I | C->U | TM9 | α-helix | - | 5 | benign | Uncertain (Inborn) |
| | T372I | C->U | ICL5 | α-helix | - | 9 | ? damaging | - |
| | T444I | C->U | Intracellular | α-helix | - | 6 | ? damaging | - |

*(Continued)*

**Table 3.** (Continued)

| Name | Mutation[1] | 2nd base[2] | Location[3] | Structure[4] | Exposure[5] | Conservation Grade[6] | Predicted Effect[7] | Clinical Significance[8] |
|------|------------|-------------|-------------|--------------|-------------|-----------------------|---------------------|--------------------------|
| | T449I | C->U | Intracellular | α-helix | - | 5 | ? damaging | - |

[1]Protein consequence of the mutation according to HGVS numbering.

[2]The second base of the residue codon for the corresponding mutation.

[3]Topological localizations of the mutations according to monoamine transporter molecular architecture (TM = Transmembrane, ECL = Extracellular loop, IM = Intramembrane, ICL = Intracellular loop). The topological information of the mature protein obtained from Uniprot.

[4]Secondary structure of the corresponding residue, calculated from the determined models of native transporters available in the AlphaFold Database.

[5]Residue exposure according to the NACSES algorithm, predicted by ConSurf server

[6]Evolutionary conservation grade of the residue predicted by ConSurf server; 1 to 9, in order of increasing conservation (1 = Variable, 5 = Average, 9 = Conserved).

[7]Variant effect predicted by Polyphen-2. Benign = predicted to be benign with high confidence;? damaging = possibly damaging, predicted to be damaging with low confidence; damaging = probably damaging: predicted to be damaging with high confidence.

[8]Based on ClinVar's April 09, 2023 release.

[9]A functional residue (exposed and highly conserved) predicted by ConSurf Server.

[10]A structural residue (buried and highly conserved) predicted by ConSurf Server.

damaging (5/25 = 20%). Regardless of their location, 22 out of 42 (52.3%) of the reverse QTY mutations were predicted to be benign.

The natural QTY or rQTY substitutions in 7 monoamine transporters reported here were not exempt from medical significance, as 11 variations found to be previously reported in ClinVar archives [43] (Tables 2 and 3). Three of the variants were benign and a total of 8 variants were linked to uncertain significance. Further analysis revealed that only one variant (p. Ile540The) among these eight was associated with a specific medical condition, namely Parkinsonism, but its significance remained uncertain (Table 2), namely, available evidence is insufficient to determine the role of the variant in disease. The variant was located on the TM11 alpha helix of DAT, with a benign predicted effect.

## Phenotypical analysis of QTY and rQTY mutation libraries

For the analysis of naturally occurring QTY and reverse QTY (rQTY) variations, we built mutation libraries by calculating the effects of all 19 amino acid substitutions possible to occur at the residue, except the wild amino acid. In particular, the median pph2_prob scores for natural QTY code variations are significantly lower when compared to other polar amino acid changes at the same residue: M = 0.635 (IQR 0.01–0.999) and M = 0.9625 (IQR 0.2255–1.0), respectively. While the median for all 19 amino acids was M = 0.947 (IQR 0.204–1.0), regardless of their polarity. These findings are also applicable for reverse QTY (rQTY) code, the median score for other nonpolar amino acid substitutions was M = 0.903 (IQR 0.167–0.99), and for the rQTY substitutions, it was only M = 0.435 (IQR 0.018–0.974). Moreover, the rQTY substitutions also showed a lower impact compared to all 19 amino acids M = 0.8955 (IQR 0.15–0.99) (S5 and S6 Figs in S1 File). The narrower Interquartile Ranges (IQR) for other polar amino acid changes could be indicative of a more definitive damaging impact on protein function, potentially due to structural limitations that have led to a less tolerated substitutions.

Variants based on populations may not be directly generalizable to artificial QTY variants as natural selection and genetic drifts in the non-Hardy-Weinberg equilibrium population introduces a sample selection bias [38]. To address this, mutation libraries are constructed for all L, I, and F amino acids in the TM regions of the seven monoamine transporters (total 861 residues), regardless of their occurrence in the population or nature. The impact of the substitutions varied (S7-S10 Figs in S1 File). Regarding the primary focus of this study, the L->Q, I-

>T, F->Y substitutions (QTY code) had a lower impact on function and structure M = 0.990 (IQR 0.612–1.0), compared to the other polar amino acids M = 0.997 (IQR 0.782–1.0), and were less damaging than the median of all substitutions M = 0.995 (IQR 0.780–1.0).

These findings are reproducible and persisted consistently in each of the seven transporters. While the QTY code targets a change in the phenotypical characteristics, such as water solubility, these results are highly promising since they indicate a lower functional disturbance, making them less prone to unintended interactions compared to other systematic substitutions.

## Evolutionary conservation studies and analysis of sensitive transmembrane (TM) domains

The phenotypical profiling indicated that the investigated TM residues are sensitive to changes. The median score of the 16,359 individual substitutions was M = 0.995 (IQR 0.780–1.0), which is further supported by mutation visualizations of the whole transporter sequence (S11 and S12 Figs in S1 File). Meanwhile, evolutionary conservation analysis also showed a diverse presence across species, such that orthologous sequences were even observed in phylum Hemichordata. Despite their early presence, the conservation grades of individual residues were not uniformly distributed. Notably, transmembrane (TM) residues of the 7 monoamine transporters were found to be more conserved compared to the motifs in the N- and C- termini (S13-S20 Figs in S1 File). This sensitivity may be attributed to the crucial role played by TM helices in maintaining the structural integrity and bilayer localization of these proteins. Conservation and the long evolutionary history of the QTY-code targeted residues is advantageous while studying their evolutionary dynamics. This is because when evolutionary forces have had a long time to exert their influence, this allows biologists to treat the dynamic as Evolutionary Stable Strategy (ESS) that corresponds a *Nash equilibrium*, and potentially without a need for an explicit solution of the underlying dynamical system [44].

Our findings raise the question of whether the opposite hydrophobic properties of the QTY code pairs drive a negative correlation in homologous sequences. Residue varieties of L and Q for each position in transporter genes indicates that the correlation coefficient $r_s(3892) = -0.0739$ while statistically significant ($p < .0001$), is indeed very small ($|r|<0.1$) and may not represent a meaningful relationship in a practical sense. The coefficients for other QTY-code pairs were calculated: +0.1915, +0.2816, +0.3721, namely for I/T, V/T, and F/Y, respectively. Contrary to the negative correlation, a highly significant positive correlation ($p<0.0001$) found in these analyses. After removing the nonlinear effect of evolutionary conservation, partial correlation coefficient between QTY-code pairs became more negative: -0.214, +0.0923, +0.165, +0.347, respectively.

Apparently, residual evolutionary pressures acted as a confounding variable to reduce the negative correlations between QTY-pairs. Hence, suggesting that rather than opposing hydrophobicity, other evolutionary pressures such as the need for specific structural motifs and functions may have played a more significant role in shaping the diversity of alpha helices. Higher F/Y coefficient is correlated with their codon presentations, where Leucine and stop codons Ochre and Amber restrict them to only encoded by remaining two codons (S4 Fig in S1 File). It is important to note that the underlying dynamics may not be monotonic, rather than they are assumed for theoretical purposes in our statistical analysis (see Methods). The correlation effect sizes ($|r|<0.5$, below large) suggest that QTY substitutions are not causing uniform changes across residues which may be caused by locational characteristics of individual residues. This can be advantageous by enabling the design of proteins with unique combinations of hydrophilic patches in specific regions.

Another significant finding is that a highly positive correlation between V and T were observed in homologous sequences, even though the V/T substitution had been not observed

in the variant analysis, since two sequential SNPs in the same codon are required. The exact evolutionary dynamics behind the phenomenon is hard to solve, however the ESS indicates that selective pressures, such as structural similarities between V and T, may counteract the low likelihood of these mutations in the long term. Although reduced, a small positive correlation ($r_s$ = 0.165) persists even after removing the confounding variable of conservation, indicating a supporting mechanism. Possibly, common ancestral amino acids that need one SNP to convert either V or T, are creating this asymmetric dynamic. Such an ancestral amino acid could be Alanine (A). In this case, C->U or G->A transition mutations are required for substitutions to Valine or Threonine, respectively. If the ancestral amino acid was Isoleucine (I); A->G or U->C mutations are needed (S4 Fig in S1 File). Considering C->T and G->A changes are more frequent than T->C and A->G changes [39, 40], Alanine could possibly cause unreciprocated V/T changes over the evolutionary course. However, this distribution is not necessarily mediated by adaptation but may result from neutral cases. Resulting in V/T divergence may have a function in molecular evolution, more detailed analysis is beneficial. Ancestral sequences are often undermined, but in this context, they may have implications for the overall equilibrium of the genetic system.

## Evolutionary perspectives and AI-enhanced computational methods

Our study concludes the presence of highly conserved residues in the transmembrane (TM) regions of the 7 monoamine transporters. Since AlphaFold relies on patterns learned from the Multiple Sequence Alignment (MSA), predictions of highly conserved regions may be challenging, whilst having less information on different variations. Even if QTY substitutions resulted in visible changes in the MSA and statistically significant differences was observed in homologous sequences, the consensus sequence remains predominantly wild type. Hence, AlphaFold can possibly incline towards conformations mirroring the consensus, that does not necessarily represent the global energy minimum of variants. Alongside the MSA utilization, AlphaFold aims to predict a single, most likely protein structure based on the given input data and does not produce a probability distribution with the Boltzmann form, contrary to the probabilistic occurrence seen in nature [45]. PolyPhen-2 and SIFT may create a pessimistic bias in mutational analyses, since substitutions leading to soluble variants (such as QTY code) are less-likely to be found in the transmembrane domains. However, it is important to acknowledge that these observations remain speculative, and further empirical evidence is required to substantiate their potential impact on predictions. It is worth noting that the previous experiments involving QTY variants that were purified and expressed, were aligned with the predictions of AlphaFold2 [15–17].

While the genetic code governs the opposing chemical properties of amino acids [16, 36, 37], we showed that evolution by natural selection can potentially pressure similar structural presentations. Thus, a comprehensive understanding of evolution, which embraces the dynamics of protein structure, is crucial. The reliance on sequence input has its advantages, however the gene deterministic assumptions may, in potential, produce intrinsic biases in predictions that favor naturally selected conformations. In this context, we highlight the need for analyses of designed proteins and conformational changes. QTY variants may be useful since they allow researchers to study proteins in a more controlled aqueous environment, with remarkable stability [15–17, 19, 20, 24].

## Possible applications and future directions of the study

Our study provides insights into the influence of amino acid substitutions in the transmembrane (TM) region of the dopamine transporter, norepinephrine transporter and serotonin

transporter, offering approaches to design diagnostics tools and therapeutics. Taking into account that QTY variants share substantial structural similarities with their transmembrane counterparts, makes them useful tools for both theoretical and practical studies. Recently developed RESENSA arrays utilize these aspects of the QTY variants for diagnostic tools [13]. Specific diagnostic tests for dopamine transporter, norepinephrine transporter and serotonin transporter have the potential to significantly improve the clinical management of neurological diseases. Moreover, these variants are useful in discovering therapeutic monoclonal antibodies. Researchers can also apply QTY-code to specific regions of a protein for drug delivery purposes and tethered agonist design. The designed QTY variants have potential applications in a wide range of research areas, ranging from antibody design to synthetic biology. Performing Molecular Dynamics simulations can further facilitate the study of functional properties of the QTY variants and may allow a more detailed interpretation of our results and effects of the mutations. Membrane localization also regulates the dynamics and Molecular Dynamics simulations in bilayer systems may provide additional insights that result from differences in water accessibility [46].

Through our extensive analysis, we uncovered the potential of systematic variations for effective protein design applications. We further provided insights into structural and phenotypical presentations and evolutionary dynamics of chemically distinct alpha helices. Accordingly, our results revealed that the applying QTY code did not alter the overall structure of the 7 monoamine transporters. Moreover, the QTY code had a notably impact on the phenotypical characteristics of the proteins under investigation.

## Methods

### Protein sequence alignments and other characteristics

The UniProt [47] website (https://www.uniprot.org) provides protein ID, entry name, description, and FASTA sequence information for each protein. The UniProt accession numbers for the studied proteins are P23975, Q01959, P31645, P54219, Q05940, Q16572, and Q6NT16, respectively. The QTY code was applied to transmembrane alpha-helices of each protein sequence, using the topological information and cellular locations of the mature proteins that were also derived from UniProt database [47]. The membrane topology and other sequence features then visualized using Protter web application (https://wlab.ethz.ch/protter/) [48]. The obtained secondary structures and sequence alignments later visualized using the 2dSS web server (http://genome.lcqb.upmc.fr/2dss/) [49].

Transmembrane helix predictions for both native transporters and their QTY variants were carried out based on a hidden Markov model using TMHMM -2.0 [50, 51]. The molecular weights (MW) and isoelectric points (pI) of the investigated proteins were calculated using the Expasy website (https://web.expasy.org/compute_pi/) [52–54].

### AlphaFold2 predictions

The structure predictions of the QTY variants were performed using the AlphaFold2 [29, 30] program, which was accessed at (https://github.com/sokrypton/ColabFold). The program was run on 2 x 20 Intel Xeon Gold 6248 cores with 384 GB of RAM and a Nvidia Volta V100 GPU, following the default instructions provided on the website. The European Bioinformatics Institute (EBI) houses over 200 million AlphaFold2-predicted structures and can be found at (https://alphafold.ebi.ac.uk).

Multiple sequence alignment (MSA) data file generated whilst AlphaFold predictions analyzed using Jalview 2.11.2.7 [55] and the alignments color coded according to hydrophobicity. Each residue in the consensus sequence is determined as the most frequent residue in each column of the alignment.

## Superposed structures

The experimentally-determined structure used in this study are SERT (PDB ID: 6VRH) [31] that were obtained from the RCSB PDB database. (https://www.rcsb.org) [32]. The native structures of seven transporters and their QTY variants were predicted using AlphaFold2 [29, 30]. The superposition of these structures was performed using PyMOL [56], which is available at (https://pymol.org/2/).

## Structure visualization

In the study, two software programs were utilized for structure visualization: PyMOL [56] (https://pymol.org/2/) and UCSF ChimeraX [57] (https://www.rbvi.ucsf.edu/chimerax/). PyMOL was used for the superposition of molecular models, whereas the molecular lipophilicity potential (MLP) maps generated with ChimeraX. Additionally, the visualization of natural mutations of the QTY and rQTY variants on Alphafold2 predicted native structures was also performed using the ChimeraX software.

## Data acquisition and variant analysis

Variants containing natural variations of QTY (L->Q, I->T, and F->Y) and reverse QTY (Q->L, T->I, and Y->F) submitted by large-scale sequencing projects obtained from the Genome Aggregation Database [58] (gnomAD v2.1.1, http://gnomad.broadinstitute.org/). Disease-associated variants accessed from the ClinVar database [43] (https://www.ncbi.nlm.nih.gov/clinvar/) resulting in a final dataset of 95 missense protein variants. The transporter topology obtained through UniProt data and *in silico* variant impact predictions obtained by Polyphen-2 [42] (http://genetics.bwh.harvard.edu/pph2/) later correlated with the missense amino acid variants. Per-residue helix and strand assignments of the studied native transporters were deduced from the models available in the AlphaFold Database [29, 30], the algorithm for Defining the Secondary Structure of Proteins (DSSP) [59] were run using UCSF ChimeraX [57] (https://www.rbvi.ucsf.edu/chimerax/). The energy cut off parameters of -0.5 kcal/mol, as recommended [59], were used for the calculations, minimum number of residues allowed in a helix or strand were also set to the default value of 3 [59]. These data were later correlated with the predicted effects of the QTY and rQTY substitutions.

## Building natural QTY and rQTY mutation libraries

PolyPhen-2 [42] (http://genetics.bwh.harvard.edu/pph2/) was used to predict the impact of the mutations on the protein function and structure. The input data for PolyPhen-2 analysis included all 19 amino acids substitutions possible to occur at the residue, which natural QTY or rQTY substitutions occurred. The Polyphen-2 algorithm considers hydrophobic potentials when predicting the effects of amino acid substitutions [42]. To address this issue, we specifically compared the effects of L->Q, I->T, and F->Y, to substitutions involving other polar amino acids: L to D, E, R, K, H, N, S, T, Y; I to D, E, R, K, H, N, S, Q, Y; and F to D, E, R, K, H, N, S, T, Q. For the reverse QTY (rQTY) variations, we compared the effects substitutions of Q->L, T->I, and Y->F, to substitutions involving other nonpolar amino acids (A, C, G, I, L, M, F, P, W, V). More than 1,800 potential variations analyzed, and the predicted effects were subsequently visualized using GNUPlot [60]. Shapiro-Wilk normality test [61] was performed and showed that the distribution of the Polyphen-2 scores departed significantly from normality (W(893) = 0.73, p-value < 0.01) and (W(798) = 0.75, p-value < 0.01), respectively. Based on this outcome, the median values with interquartile ranges (IQR) were primarily used to summarize the phenotypic scores. Statistical calculations were performed using R (The R

Foundation for Statistical Computing, Vienna, Austria), version 4.3.1 (https://www.r-project.org/) [62, 63].

To overcome potential sampling biases, we used Polyphen-2 [42] to predict the effects of all 19 amino acids substitutions at the residue of L, I, V, F amino acids in the TM α-helices of the proteins, regardless of their occurrence in the population or nature. The predicted effects of 16359 variations (that corresponds to 861 residues) were used as input data for the 3D plots generated using GNUPlot [60]. The median values, interquartile ranges (IQR) and average values were calculated. L, I, V, F -> Q, T, Y substitutions compared with other amino 19 acid substitutions at the residue.

## Evolutionary conservation profiles and analysis of sensitive domains

Mutation visualizations for dopamine transporter, norepinephrine transporter and serotonin transporter were generated according to predictions for whole proteins that were accessed from SIFT [64] (https://sift.bii.a-star.edu.sg/). ConSurf server [65–70] (https://consurf.tau.ac.il/) used for calculating evolutionary conservation profiles. The server ran with AlphaFold2 predicted native structures that were also used for RMSD calculations. These structures were later complemented with SEQRES records that were derived from Uniprot in FASTA format. To translate and add the amino acid sequences to the.pdb files in the correct SEQRES format, visual basic for applications (VBA) scripting was utilized.

The conservation scores were computed using the Bayesian method, with the amino acid substitution model chosen based on the best fit. The default parameters were employed for homologues search, phylogeny, and conservation scores. The evolutionary conservation grades of each residue were visualized using the UCSF ChimeraX [57] (https://www.rbvi.ucsf.edu/chimerax/). The conservation grades and residue exposure data obtained from the ConSurf server were complemented with secondary structure information and transporter topology. Homologous sequences obtained from UniRef90, that is built by clustering UniRef100 sequences the MMseqs2 algorithm [65, 71]. Phylogenetic trees constructed by ConSurf using neighbor joining with ML distance, that were later visualized using Jalview 2.11.2.7 [55].

The QTY-code pairs in the ConSurf MSA were analyzed via different statistical measures. The variations were most likely not in Hardy-Weinberg equilibrium as they are assumed to do so in our statistical analysis. Our data is not in time continuous; hence it is hard to assume if the evolutionary dynamics was uniformly aggregate monotone [44]. However, the Hardy-Weinberg equilibrium and ESS assumptions serve as a useful theoretical baseline for understanding how genetic variation can change due to evolutionary forces [44]. Since the variables did not exhibit a normal distribution according to the Shapiro-Wilk tests [61], we utilized the Wilcoxon signed-rank test [72] to assess the significance of differences in the non-parametric data. Because of the violation of the assumptions of linearity and normality for the variables, nonparametric analyses were performed, and bootstrap was used in the analyses to calculate confidence intervals [73]. The Spearman's rank correlation coefficient (Spearman's ρ) was used to measure the monotonic association between two variables (L-Q, I-T, V-T, F-Y).

Evolutionary conservation is hypothesized as a confounding variable. For the conservation estimations, ConSurf scores are used instead of Shannon entropy or other entropy-based methods. ConSurf uses a phylogenetic approach, considering evolutionary history and phylogenetic relationships between sequences, which is better suited for this study. The partial correlation coefficients were calculated to assess the correlation after removing the nonlinear effects of the confounding variable of the evolutionary conservation. Since homoscedasticity assumptions may not met by our non-parametric data, the Spearman partial rank-order correlation coefficients are used as the multivariate analysis [74]. Moreover, the variables might

have a non-linear relationship and contain outliners, which Pearson's coefficients are more sensitive of [75]. All statistical calculations were performed using R (The R Foundation for Statistical Computing, Vienna, Austria), version 4.3.1 (https://www.r-project.org/) [62, 63, 73, 74].

## Supporting information

**S1 File.**
(DOCX)

## Acknowledgments

This work was partly funded by 511 Therapeutics. We also thank Dorrie Langsley for English editing.

## Additional statements

Shuguang Zhang is also a co-founder and Scientific Advisor for 3DMatrix Co Ltd in Japan that produces self-assembling peptide hydrogels for accelerating wound healing for surgical and dental wound healing applications.

**Correspondence** and request for materials should be address to Shuguang Zhang, Shuguang@MIT.EDU

## Author Contributions

**Conceptualization:** Shuguang Zhang.

**Data curation:** Taner Karagöl, Alper Karagöl.

**Formal analysis:** Taner Karagöl, Alper Karagöl.

**Investigation:** Taner Karagöl, Alper Karagöl.

**Methodology:** Taner Karagöl, Alper Karagöl.

**Project administration:** Shuguang Zhang.

**Software:** Taner Karagöl, Alper Karagöl.

**Supervision:** Shuguang Zhang.

**Validation:** Taner Karagöl, Alper Karagöl.

**Visualization:** Taner Karagöl, Alper Karagöl.

**Writing – original draft:** Taner Karagöl, Alper Karagöl.

**Writing – review & editing:** Taner Karagöl, Alper Karagöl, Shuguang Zhang.

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
