## [Decision Letter · Decision Letter 0]

19 Dec 2023

PONE-D-23-35512Structural bioinformatics studies of serotonin, dopamine and norepinephrine transporters and their AlphaFold2 predicted water-soluble QTY variants and uncovering the natural mutations of L->Q, I->T, F->Y and Q->L, T->I and Y->FPLOS ONE

Dear Dr. Zhang,

Thank you for submitting your manuscript to PLOS ONE. After careful consideration, we feel that it has merit but does not fully meet PLOS ONE’s publication criteria as it currently stands. Therefore, we invite you to submit a revised version of the manuscript that addresses the points raised during the review process.

We look forward to receiving your revised manuscript.

Kind regards,

Ahmed E. Abdel Moneim

Academic Editor

PLOS ONE

Journal Requirements:

3. We note that you have a patent relating to material pertinent to this article. Please provide an amended statement of Competing Interests to declare this patent (with details including name and number), along with any other relevant declarations relating to employment, consultancy, patents, products in development or modified products etc. Please confirm that this does not alter your adherence to all PLOS ONE policies on sharing data and materials, as detailed online in our guide for authors http://journals.plos.org/plosone/s/competing-interests by including the following statement: "This does not alter our adherence to  PLOS ONE policies on sharing data and materials.” If there are restrictions on sharing of data and/or materials, please state these. Please note that we cannot proceed with consideration of your article until this information has been declared.

Thank you for uploading your study's underlying data set. Unfortunately, the repository you have noted in your Data Availability statement does not qualify as an acceptable data repository according to PLOS's standards.

Reviewers' comments:

Reviewer's Responses to Questions

**Comments to the Author**

1. Is the manuscript technically sound, and do the data support the conclusions?

Reviewer #1: Yes

Reviewer #2: Yes

Reviewer #3: Yes

2. Has the statistical analysis been performed appropriately and rigorously? 

Reviewer #1: Yes

Reviewer #2: Yes

Reviewer #3: Yes

3. Have the authors made all data underlying the findings in their manuscript fully available?

Reviewer #1: Yes

Reviewer #2: Yes

Reviewer #3: Yes

4. Is the manuscript presented in an intelligible fashion and written in standard English?

Reviewer #1: Yes

Reviewer #2: Yes

Reviewer #3: Yes

5. Review Comments to the Author

Reviewer #1: Comments: Minor Revision

This study applies a water solubilization design method called QTY code to monoamine transporters, computationally characterizing some properties of the designed proteins and investigating the relationship between natural mutations in these monoamine transporters and the QTY code. These findings are intriguing and hold some significance for the study on monoamine transporters and the QTY code; however, there are some issues in the manuscript should be addressed by the authors.

Specific comments:

1. Abstract：While their hydrophobic surfaces significantly reduced, this change did not cost a structural alteration.

This statement tends to be too assertive. However, after numerous sequence variations, there must be structural changes, to varying degrees.

2. Abstract：Accompanied by similarities of QTY-code pairs, our study unrevealed multiple QTY and reverse QTY variations in genomic databases.

Please provide a more explicit explanation regarding the specific aspects of similarity. Additionally, the use of the term 'QTY-code pair' in the abstract is also confusing.

3. The structure of Introduction section needs improvement. The current manuscript presents introductions to various aspects (transporter, QTY code, transporter mutation) in a somewhat confusing manner, with much interweaving. This may make it challenging for readers to quickly and clearly grasp the background knowledge on these topics.

4. The order of figures needs to be revised, as Figure S19 appears earliest and should not be the last supplementary figure.

5. Page 4: The QTY code substitutes four hydrophobic amino acids (Leucine, Isoleucine, Valine, and Phenylalanine) with hydrophilic amino acids (Glutamine = Q, Threonine = T, and Tyrosine = Y)

The content related to the QTY code has already been introduced earlier; no need to repeat. The meanings of single letters also do not need to be redundantly explained.

6. Page 4: The highly predicted Local Distance Difference Test (pLDDT) scores of the predictions are reassuring. Indicating greater confidence in predicted protein structures [33].

This sentence needs to be rewritten.

7. In Page 4, it is mentioned that the pLDDT score is used to assess the quality of AlphaFold2 predictions. It is quite perplexing that the authors have not provided the pLDDT scores for the QTY variants and their distribution in the structures.

8. In Page 4, a significant portion is dedicated to introduce hydrophilic and hydrophobic alpha-helices. It is recommended to place this information after the results and reduce its length.

9. Table 2 consists of only one row and does not need to be presented as a separate table.

Reviewer #2: This manuscript bioinformatically studies the structure and properties of monoamine transporters and their QTY designed variants based on AlphaFold2 and tools such as ClinVar and PolyPhen-2.

The author took a step forward from his previous QTY designed membrane proteins to include more genomic analysis and evolutionary conservation of specific sites related to Q, T, Y and L, I, F, so as to provide more information regarding phenotypical impacts from these substitutions, as well as providing more guidance in subsequent designs.

The study is very interesting and is suitable for publication in PLOS One following minor revision.

1. Please check the spelling and formatting of the entire manuscript. Such as, page 8, “sequance” would be “sequence”. Page 13, the format of reference 13 is inappropriate.

2. Sequence in Figure. 1 is barely visible.

3. The current research on QTY in the introduction is not comprehensive, and relevant research background needs to be supplemented. Please note that all correctly used papers should be cited.

4. There could be more discussion about whether 11 variations found to be previously reported in ClinVar archives could be discussed. This part may have referential significance for the treatment of related diseases.

Reviewer #3: The paper is well-written, and the problem this study is dedicated to is important, as justified in the introduction. The authors provide an additional GitHub repository with the structural data used in this work, making it reproducible. The evolution study is the most interesting, however, the structural part can be improved with MD simulations. Nevertheless, the authors mention all the shortcomings that can be connected to the work, making the paper of good quality. Overall, I recommend this paper for publication.

Suggestions:

"Despite this effort, structural information on the monoamine transporters is still limited and many aspects of their molecular mechanisms have remained unknown" - To understand the mechanics behind these transporters and the more realistic effects of QTY mutations on the protein's structure and function, it is worth investigating the systems in dynamics. If you cannot conduct MD, please at least add to the text a discussion of how MD simulations can influence your results and findings. However, for further studies, I do recommend including MD simulations. You can set them up in the following way: 1) native protein in the membrane, 2) native protein in solute, 3) QTY protein in membrane, 4) QTY protein in solute for at least two different proteins that perform best in the rigid models comparisons. Next, evaluate the stability and behavior of all these 8 systems (4 setups x 2 proteins). Calculate the number of hydrophobic and hydrophilic contacts for each system and conclude how hydrophobicity changes from native to QTY systems.

6. PLOS authors have the option to publish the peer review history of their article (what does this mean?). If published, this will include your full peer review and any attached files.

Reviewer #1: No

Reviewer #2: No

Reviewer #3: No

---

## [Author Response · Author response to Decision Letter 0]

2 Feb 2024

Responses to Reviewers

Reviewer #1: Comments: Minor Revision 

This study applies a water solubilization design method called QTY code to monoamine transporters, computationally characterizing some properties of the designed proteins and investigating the relationship between natural mutations in these monoamine transporters and the QTY code. These findings are intriguing and hold some significance for the study on monoamine transporters and the QTY code; however, there are some issues in the manuscript should be addressed by the authors. 

Specific comments: 

1. Abstract： 

While their hydrophobic surfaces significantly reduced, this change did not cost a structural alteration. 

This statement tends to be too assertive. However, after numerous sequence variations, there must be structural changes, to varying degrees.

We appreciate Reviewer #1’s positive feedback. We have revised the statement about the structural alterations and provided clearer information. “While their hydrophobic surfaces significantly reduced, this change resulted in a minimal a structural alteration.”

2. Abstract：

Accompanied by similarities of QTY-code pairs, our study unrevealed multiple QTY and reverse QTY variations in genomic databases. 

Please provide a more explicit explanation regarding the specific aspects of similarity. Additionally, the use of the term 'QTY-code pair' in the abstract is also confusing.

We have provided a more explicit explanation structural similarity among QTY-code pairs and addressed the confusion surrounding the term ‘QTY-code pair’ in the abstract. ” Accompanied by the structural similarities of substituted amino acids in the context of 1.5Å electron density maps, our study revealed multiple QTY and reverse QTY variations in genomic databases.”

3. The structure of Introduction section needs improvement. The current manuscript presents introductions to various aspects (transporter, QTY code, transporter mutation) in a somewhat confusing manner, with much interweaving. This may make it challenging for readers to quickly and clearly grasp the background knowledge on these topics.

We have restructured the introduction section to provide a clearer and more organized presentation, by rearranging content on the significance of the studied transporters before the background information on the QTY code.

4. The order of figures needs to be revised, as Figure S19 appears earliest and should not be the last supplementary figure.

The order of figures has been revised, Figure S19 is rearranged as Figure S1.

5. Page 4: The QTY code substitutes four hydrophobic amino acids (Leucine, Isoleucine, Valine, and Phenylalanine) with hydrophilic amino acids (Glutamine = Q, Threonine = T, and Tyrosine = Y) The content related to the QTY code has already been introduced earlier; no need to repeat. The meanings of single letters also do not need to be redundantly explained.

The content related to the QTY code has been streamlined, eliminating unnecessary repetition.

6. Page 4: The highly predicted Local Distance Difference Test (pLDDT) scores of the predictions are reassuring. Indicating greater confidence in predicted protein structures [33]. This sentence needs to be rewritten.

The sentence regarding the Local Distance Difference Test (pLDDT) scores has been rewritten for clarity. “High predicted Local Distance Difference Test (pLDDT) scores for the predictions also indicated a high degree of confidence in their accuracy, especially in helical regions (Figure S2). However, it's important to note that pLDDT scores utilize the information available in the training data, which may not perfectly reflect specific biological contexts”.

7. In Page 4, it is mentioned that the pLDDT score is used to assess the quality of AlphaFold2 predictions. It is quite perplexing that the authors have not provided the pLDDT scores for the QTY variants and their distribution in the structures.

We have acknowledged the omission of pLDDT scores for QTY variants and their distribution in the structures and included this information in the revised supplementary information (Figure S3).

8. In Page 4, a significant portion is dedicated to introduce hydrophilic and hydrophobic alpha-helices. It is recommended to place this information after the results and reduce its length.

The paragraph on hydrophilic and hydrophobic alpha-helices in Page 4 has been reduced.” Nature has evolved three chemically distinct types of alpha-helices [14,34,35]. The hydrophilic alpha-helices contain mostly polar amino acids D, E, N, Q, K, R, S, T, and Y [14], as found in water-soluble proteins. On the other hand, hydrophobic alpha-helices are present in transmembrane proteins and contain mostly hydrophobic amino acids [14]. Amphiphilic alpha-helices contain both hydrophobic and hydrophilic amino acid residues.”

9. Table 2 consists of only one row and does not need to be presented as a separate table.

Acknowledging the single-row content, we have integrated the information into the text and deleted Table 2.

Reviewer #2: 

This manuscript bioinformatically studies the structure and properties of monoamine transporters and their QTY designed variants based on AlphaFold2 and tools such as ClinVar and PolyPhen-2. The author took a step forward from his previous QTY designed membrane proteins to include more genomic analysis and evolutionary conservation of specific sites related to Q, T, Y and L, I, F, so as to provide more information regarding phenotypical impacts from these substitutions, as well as providing more guidance in subsequent designs. The study is very interesting and is suitable for publication in PLOS One following minor revision. 

1. Please check the spelling and formatting of the entire manuscript. 

Such as, page 8, “sequance” would be “sequence”. 

Page 13, the format of reference 13 is inappropriate.

Spelling and formatting issues have been addressed throughout the manuscript.

2. Sequence in Figure. 1 is barely visible.

3. The current research on QTY in the introduction is not comprehensive, and relevant research background needs to be supplemented. Please note that all correctly used papers should be cited.

The introduction section has been supplemented with more comprehensive background information on the previous QTY research.” The QTY code was previously applied to design a range of detergent-free chemokine receptors and cytokine receptors, and the expressed and purified water-soluble variants exhibited the predicted characteristics and maintained their ligand-binding activity [13,14,15,17,18]. After its release in July 2021, we started using AlphaFold2 to make QTY variant protein structure predictions and achieved improved results, significantly faster [19,20,21,22,23]. In which, the QTY variants of transporter proteins, including potassium channels and glucose transporters, showed a remarkable similarity with their native counterparts. [20,21,22]. Concurrently, reverse QTY variants of Human Serum Albumin successfully designed and demonstrated efficacy in facilitating the release of anti-tumor drugs in mice [16].”

4. There could be more discussion about whether 11 variations found to be previously reported in ClinVar archives could be discussed. This part may have referential significance for the treatment of related diseases.

A more in-depth discussion about the relevant variation found in ClinVar archives has been added to the manuscript. “The natural QTY or rQTY substitutions in 7 monoamine transporters reported here were not exempt from medical significance, as 11 variations found to be previously reported in ClinVar archives [43] (Table 2, Table 3). Three of the variants were benign and a total of 8 variants were linked to uncertain significance. Further analysis revealed that only one variant (p.Ile540The) among these eight was associated with a specific medical condition, namely Parkinsonism, but its significance remained uncertain. (Table 2). Meaning, available evidence is insufficient to determine the role of the variant in disease. The variant was located on the TM11 alpha helix of DAT, with a benign predicted effect.”

Reviewer #3:

The paper is well-written, and the problem this study is dedicated to is important, as justified in the introduction. The authors provide an additional GitHub repository with the structural data used in this work, making it reproducible. The evolution study is the most interesting, however, the structural part can be improved with MD simulations. Nevertheless, the authors mention all the shortcomings that can be connected to the work, making the paper of good quality. Overall, I recommend this paper for publication.

Suggestions: 

"Despite this effort, structural information on the monoamine transporters is still limited and many aspects of their molecular mechanisms have remained unknown" - To understand the mechanics behind these transporters and the more realistic effects of QTY mutations on the protein's structure and function, it is worth investigating the systems in dynamics. If you cannot conduct MD, please at least add to the text a discussion of how MD simulations can influence your results and findings. However, for further studies, I do recommend including MD simulations. You can set them up in the following way: 

1) native protein in the membrane, 2) native protein in solute, 3) QTY protein in membrane, 4) QTY protein in solute for at least two different proteins that perform best in the rigid models comparisons. Next, evaluate the stability and behavior of all these 8 systems (4 setups x 2 proteins). Calculate the number of hydrophobic and hydrophilic contacts for each system and conclude how hydrophobicity changes from native to QTY systems.

We appreciate your positive feedback and suggestion for including MD simulations. While MD simulations were not conducted in this study, we have added a discussion on how MD simulations could influence the results and findings. Additionally, we have included recommendations for future studies, emphasizing the importance of MD simulations in investigating the dynamics of native and QTY-mutated proteins.” The designed QTY variants have potential applications in a wide range of research areas, ranging from antibody design to synthetic biology. Performing Molecular Dynamics simulations can further facilitate the study of functional properties of the QTY variants and may allow a more detailed interpretation of our results and effects of the mutations. Membrane localization also regulates the dynamics and Molecular Dynamics simulations in bilayer systems may provide additional insights that result from differences in water accessibility. [46].”

---

## [Decision Letter · Decision Letter 1]

27 Feb 2024

Structural bioinformatics studies of serotonin, dopamine and norepinephrinetransporters and their AlphaFold2 predicted water-soluble QTY variants and uncovering the natural mutations of L->Q, I->T, F->Y and Q->L, T->I and Y->F

PONE-D-23-35512R1

Dear Dr. Zhang,

We’re pleased to inform you that your manuscript has been judged scientifically suitable for publication and will be formally accepted for publication once it meets all outstanding technical requirements.

Kind regards,

Ahmed E. Abdel Moneim

Academic Editor

PLOS ONE

Additional Editor Comments (optional):

Reviewers' comments:

Reviewer's Responses to Questions

**Comments to the Author**

1. If the authors have adequately addressed your comments raised in a previous round of review and you feel that this manuscript is now acceptable for publication, you may indicate that here to bypass the “Comments to the Author” section, enter your conflict of interest statement in the “Confidential to Editor” section, and submit your "Accept" recommendation.

Reviewer #1: All comments have been addressed

Reviewer #2: All comments have been addressed

2. Is the manuscript technically sound, and do the data support the conclusions?

Reviewer #1: Yes

Reviewer #2: Yes

3. Has the statistical analysis been performed appropriately and rigorously? 

Reviewer #1: Yes

Reviewer #2: Yes

4. Have the authors made all data underlying the findings in their manuscript fully available?

Reviewer #1: Yes

Reviewer #2: Yes

5. Is the manuscript presented in an intelligible fashion and written in standard English?

Reviewer #1: Yes

Reviewer #2: Yes

6. Review Comments to the Author

Reviewer #1: The authors have addressed my concerns in this round of revision. I recommend this manuscript for publication.

Reviewer #2: The authors have addressed all my comments and thus I recommend the manuscript for publication in current state.

7. PLOS authors have the option to publish the peer review history of their article (what does this mean?). If published, this will include your full peer review and any attached files.

Reviewer #1: No

Reviewer #2: No

---

## [Editor Report · Acceptance letter]

13 Mar 2024

PONE-D-23-35512R1 

PLOS ONE

Dear Dr. Zhang, 

I'm pleased to inform you that your manuscript has been deemed suitable for publication in PLOS ONE. Congratulations! Your manuscript is now being handed over to our production team.

Kind regards, 

on behalf of

Dr. Ahmed E. Abdel Moneim 

Academic Editor

PLOS ONE